# Short Beak and Dwarfism Syndrome in Ducks in Poland Caused by Novel Goose Parvovirus

**DOI:** 10.3390/ani10122397

**Published:** 2020-12-15

**Authors:** Anna Karolina Matczuk, Monika Chmielewska-Władyka, Magdalena Siedlecka, Karolina Julia Bednarek, Alina Wieliczko

**Affiliations:** 1Division of Microbiology, Department of Pathology, Faculty of Veterinary Medicine, Wroclaw University of Environmental and Life Sciences, Norwida 31, 51-375 Wroclaw, Poland; 2Department of Epizootiology with Clinic of Birds and Exotic Animals, Faculty of Veterinary Medicine, Wroclaw University of Environmental and Life Sciences, Grunwaldzki 45, 50-366 Wroclaw, Poland; monika.chmielewska-wladyka@upwr.edu.pl (M.C.-W.); magdalena.siedlecka@upwr.edu.pl (M.S.); alina.wieliczko@upwr.edu.pl (A.W.); 3AGRO-VET Veterinary Laboratory, Kuropatwia 2, 51-419 Wroclaw, Poland; pcr@agrovet.pl

**Keywords:** novel goose parvovirus, SBDS, short beak and dwarfism syndrome, Pekin duck, duck circovirus

## Abstract

**Simple Summary:**

Short beak and dwarfism syndrome (SBDS) is an emerging disease of Pekin ducks, which is caused by infection with a genetic variant of goose parvovirus. Since 2015, outbreaks have occurred in many parts of China, causing economic losses due to high mortality and morbidity. In 2019, SBDS was observed for the first time in Poland in the farms of Pekin ducks, where the birds were found to be infected with novel goose parvovirus (nGPV). In this study, the complete coding regions of Polish nGPV were sequenced. This is the first report of the new GPV variant detected in Poland, and to our knowledge, the first documented outbreak of nGPV in Pekin ducks in Europe.

**Abstract:**

Short beak and dwarfism syndrome (SBDS), which was previously identified only in mule ducks, is now an emerging disease of Pekin ducks in China and Egypt. The disease is caused by the infection of ducks with a genetic variant of goose parvovirus—novel goose parvovirus (nGPV). In 2019, SBDS was observed for the first time in Poland in eight farms of Pekin ducks. Birds in the affected flock were found to show growth retardation and beak atrophy with tongue protrusions. Morbidity ranged between 15% and 40% (in one flock), while the mortality rate was 4–6%. Co-infection with duck circovirus, a known immunosuppressive agent, was observed in 85.7% of ducks. The complete coding regions of four isolates were sequenced and submitted to GenBank. The phylogenetic analysis revealed a close relationship of Polish viral sequences with the Chinese nGPV. Genomic sequence alignments showed 98.57–99.28% identity with the nGPV sequences obtained in China, and 96.42% identity with the classical GPV (cGPV; Derzsy’s disease). The rate of amino acid mutations in comparison to cGPV and Chinese nGPV was higher in the Rep protein than in the Vp1 protein. To our knowledge, this is the first report of nGPV infection in Pekin ducks in Poland and Europe. It should be emphasized that monitoring and sequencing of waterfowl parvoviruses is important for tracking the viral genetic changes that enable adaptation to new species of waterbirds.

## 1. Introduction

Short beak and dwarfism syndrome (SBDS), which is also known as beak atrophy and dwarfism syndrome, is an emerging disease of Pekin ducks in many countries. In the 1970s, a similar disease was observed in France in the flocks of mule ducks [1]. In Poland, SBDS was first observed in mule duck flocks in 1995 [2]. Since 2015, many outbreaks were recorded in Pekin ducks in multiple locations in China, which have caused enormous economic losses due to high morbidity and reduction in the weight and size of the birds [3,4,5,6]. Furthermore, SBDS was recently observed in Egypt in mule and Pekin ducks [7].

In China, the ducks affected with SBDS showed strong growth retardation with beak atrophy, enteritis, and paralysis. The mortality rate ranged from 2% to 6% in the affected flocks, but the losses were especially linked to high morbidity which was on average 20% or higher in some regions [3,6,8].

The etiological agent of SBDS in Pekin and mule ducks is novel goose parvovirus (nGPV), which is genetically very similar to the closely related classical goose parvovirus (cGPV), the causative agent of Derzsy’s disease observed in geese. The phylogenetic analysis showed that the Chinese nGPV strains isolated from Pekin ducks and mule ducks with SDBS shared 90.8–94.6% of nucleotide identity with cGPV isolates [9].

Muscovy and mule ducks (a cross-breed of Muscovy and Pekin duck) can also be infected with Muscovy duck parvovirus. All three viruses belong to the same species *Anseriform dependoparvovirus* 1 of genus *Dependoparvovirus* and family *Parvovirinae* [10]. The viruses belonging to *Anseriform dependoparvovirus* 1 have a linear, single-stranded DNA genome of about 5-kbp length [11]. The genome contains two major open reading frames (ORFs): a nonstructural ORF *rep* encoding regulatory protein Rep and a structural ORF encoding capsid proteins Vp1, Vp2, and Vp3. The translation of Vp2 is probably initiated by the unusual start codon downstream of the Vp1 start codon, while Vp3 is translated from the same mRNA subset by the leaky scanning mechanism. Thus, the Vp2 and Vp3 proteins share the same carboxy-terminal portion with Vp1. The Vp3 protein is the most abundantly expressed and variable of the three capsid proteins, and is also responsible for the production of induction-neutralizing antibodies and providing protective immunity to waterfowl [12]. Additionally, there are two inverted terminal repeats at both ends of the genome, which can fold on themselves to form a palindromic hairpin structure [11].

Some authors suggest that two viruses—nGPV and duck circovirus (DuCV)—may be involved in the pathology of SBDS. An investigation in China on SBDS-infected Pekin ducks showed that among the birds positive for nGPV 72.48% were co-infected with DuCV. Therefore, the authors suggested that co-infection of nGPV and DuCV can be important for the development of clinical signs of SBDS [4]. DuCV is a member of the *Circovirus* genus within the *Circoviridae* family. It is a nonenveloped, single-stranded DNA virus, with a genome of approximately 2 kbp in size. The virus has been found to spread worldwide in domestic and wild ducks, and associated with feather disorders and immunosuppressive effects since it was first reported in Germany in 2003 [13,14,15]. In recent years, Poland has become a leader in poultry production. The domestic poultry industry is currently dominated by chickens and turkeys, but waterfowl are also gaining importance for economic reasons. There is a substantial increase in the number of fattening ducks. In 2018, there were 13 million hatched ducklings, while in 2019 there were 15.4 million [16]. Along with the intensification of waterfowl production, the epidemiological threat also increases. Therefore, it is highly important to monitor for emerging viruses in water poultry.

At the end of 2019, veterinarians in Poland reported the presence of symptoms resembling that of SBDS in the flocks of Pekin ducks, such as short beak, tongue dysplasia, and growth retardation.

Therefore, this study aimed to check if the clinical disease observed in Pekin ducks in Poland was due to nGPV infection. Due to the lack of sequences of nGPV from outside of China, we compared and characterized the Polish sequences with those available in GenBank. We also tested for the co-infection with DuCV. To our knowledge, there is no nGPV sequence of European origin in GenBank. In this study, we sequenced four complete coding regions of nGPV isolated from the Pekin duck farms in Poland and also performed molecular characterization of those sequences.

## 2. Materials and Methods

The study was carried out on clinical samples taken from eight flocks of Pekin ducks (Table 1), which were presenting signs of SBDS and aged 3–6 weeks, in Polish farms (Western and Central Poland). The samples obtained from the affected flocks were either cloacal swabs or pooled internal organs (liver, spleen). The veterinarians responsible for the flocks, and according to Polish animal experiments low collected the samples, such samples do not require permission from Local Ethical Committee.

The viral DNA was extracted using Syngen Viral Mini Kit Plus (Syngen Biotech, Wroclaw, Poland) and subjected to three different tests. To target the GPV, real-time polymerase chain reaction (PCR) [17] was performed with probe quantitative PCR Master Mix (2×) (Eurx, Gdańsk, Poland). In addition, a PCR to detect DuCV was performed as described in a previous study [18].

To determine the nearly complete sequence of nGPV, samples from four farms (numbers 5–8, Table 1) were subjected to genome sequencing (Sanger sequencing) using primers described previously [19], with modifications. Primers P3 to P6, forward and reverse, were used as described, while for P2 fragment we designed a primer set by amplifying an 1146-nt fragment. The sequences in the primer set were as follows: P2nF 5′-GTTTAGTTCATTCGTTACTC-3′ and P2nR 5′-CTCATTAGTCCAGTTAACG-3′. All standard PCRs were carried out with Gene-On One-Phusion polymerase (Abo, Poland), dNTPs mix (Fermentas, Abo, Poland), and relevant primers (Genomed, Warsaw, Poland). The PCR products were visualized on a 1.5% agarose gel. The fragments subjected to sequencing were cut out from gels and cleaned with GeneClean II kit (MP Biochemicals, Warsaw, Poland). All sequencing was done bidirectionally, using forward and reverse primers (Genomed, Warsaw, Poland). Five fragments obtained were composed manually after alignment in MEGA7 software [20]. For eight samples indicated in Table 1, only P5 fragment was amplified and sequenced in Genomed (Warsaw, Poland). All the sequences determined in this study were deposited in GenBank under accession numbers MW147177–MW147183 and MW 147187.

The sequence alignments and homology comparison were performed using the Clustal W method implemented in MEGA7 software [20]. Phylogenetic analysis was carried out based on nucleotide sequence comparison—complete coding region of the four sequences was compared with other full genome or nearly full genome sequences available in GenBank. A second phylogenetic analysis was performed for the P5 fragment (covering partial ORFs for Vp1 and Vp2) and other corresponding sequences available in the GenBank database, using MEGA7 software. Phylogenetic trees were generated for complete coding region sequence and P5 fragment using the neighbor-joining method as implemented in MEGA7 software. The robustness of the trees was evaluated by bootstrapping multiple sequence alignments (1000 sets).

## 3. Results

The first clinical signs such as difficulties in moving and feeding were observed in ducks at 7–10 days of life. After 3–4 weeks, the typical signs of SBDS were noticed, including shortening of the beak and protruding and dropping tongue (Figure 1). Later, the flock was found to be visibly differentiated in size, and the affected birds showed strong growth retardation, feather losses, lameness, and watery diarrhea. Morbidity ranged between 15% and 40% (in one flock), while the mortality rate was 4–6%. A higher mortality rate was observed in the flocks where GPV was detected as early as the third week of age.

All the eight flocks investigated were identified to be positive for the presence of *Anseriform dependoparvovirus* 1 by real-time PCR. The Ct values ranged from 23 to 32 (Table 1). DuCV sequence was detected in six out of seven tested samples (85.7%) (lack of DNA for one sample).

The assembled sequences P2–P6 of the samples PL156.2019, PL157.2019, PL165.2019, and Pl53.2020 were 4330-nt long and comprised the protein-coding part of the genome (85% nucleotide coverage of the whole-genome sequence of reference nGPV). We were unable to obtain the PCR products for the 3′ and 5′ ends of the genome. Within the four sequences, the nucleotide identity with each other was between 99.8% and 100%. In Blastn, the nucleotide identity of nGPV sequences with those obtained in China ranged from 98.57% to 99.28%. The nucleotide identity of Polish nGPV with cGPV (GenBank number: U25749.1) and with the sequence from ornamental ducks obtained in 2016 from Poland (GenBank number: KU684472), which is the only available whole-genome sequence of GPV from the country, was 96.42% and 97.17% respectively. The phylogenetic analysis of this coding region revealed closed relationship to Chinese nGPV (Figure 2A).

For other samples, we chose to amplify the P5 fragment for phylogenetic analysis, as we noticed high nucleotide substitution in this fragment in the sequences analyzed for the whole coding region. Moreover, other researchers have considered partial *vp1* gene sequencing for phylogenetic analysis. From other samples, we obtained a good-quality sequence for GPV P5 fragment with a length of 1037 nt, which comprises a part of the viral capsid gene (*vp1*, *vp2*, *vp3*, coding nucleotide sequence of 371 aa out of 732 aa of Vp1 protein). We also included shorter sequences (~450 bp) available for other GPV strains described in a previous study [1] for Hoekstra vaccine strain. Phylogenetic analysis of this fragment revealed that recent Polish sequences isolated from Pekin ducks clustered with Chinese nGPV sequences, but formed a distinct clade. In addition, the D146/02 strain isolated from mule ducks in France in 2002 from an SBDS-affected flock was found to belong to the nGPV clade. The sequences of this clade were clearly distinct from the cGPV fragments isolated from geese and ducks in Poland in previous years (Figure 2B).

An amino acid analysis was performed in MEGA7 software. The amino acid sequences of Rep and Vp1 were compared to those of reference cGPV, Polish cGPV, and Chinese nGPV (GenBank numbers indicated in Table 2). A unique amino acid mutation in Vp1 protein (Lys > 92 > Arg) and eight or nine unique amino acid mutations in Rep protein were identified in the Polish nGPV strains isolated in this study. The Polish sequences differed in four or five and seven or eight amino acid mutations compared to the Vp1 sequences determined in Chinese nGPV and cGPV, respectively. In the case of Rep protein, eight or nine unique amino acid mu tations were noted, which were also different from the Polish and Chinese nGPV Rep sequences. When compared to cGPV sequences, 21 or 22 amino acid mutations were found (Table 3).

## 4. Discussion

Until the emergence of the new virus in China in 2015, SBDS was only considered as a disease of mule ducks (Pekin and Muscovy duck hybrid), which were infected with GPV causing Derzsy’s disease. SBDS is a long-known disease and has been observed since the 1970s in France and other west European countries in mule duck flocks [1]. In Poland, the disease in mule ducks was first identified in 1995. GPV was isolated and the virus was found to cross-react with the sera positive for Derzsy’s disease-causing virus [2]. Genetic analysis was not widely available in the past, and because the etiological agent of SBDS cross-reacted with the sera against Derzsy’s disease, the diseases caused by it in mule ducks were classified as GPV infections. Vaccination of mule ducks with the GPV vaccine reduced the incidence of SBDS in those flocks.

In this study, we describe the outbreak of SBDS observed in eight Pekin duck flocks in Poland. The phylogenetic analysis of the nearly complete GPV genome isolated in this study clearly showed that the etiological agent of SBDS that was observed among Pekin ducks in Poland in 2019 was nGPV. To our knowledge, this is the first report of the coding region in the genome of nGPV isolated from Pekin ducks from outside of Asia; in particular, no sequences are available from Europe. Recently, SBDS was observed in Pekin and mule ducks in Egypt, and infection by nGPV was confirmed with partial sequencing of the *vp1* gene [7].

The morbidity and mortality rate detected in the Polish flocks were comparable to those resulting from the outbreaks in China [3,6,8]. A significant loss to producers was related to growth retardation, as many birds from the affected flocks did not reach the appropriate body weight. Their final weight was only approximately 0.5 kg, and such birds were sent to slaughterhouse.

To fight SBDS in the affected farms, in the subsequent production cycle, Pekin ducks were vaccinated in the first week of life with live attenuated GPV vaccine used for geese and mulard ducks (Palmivax). SBDS did not appear in the subsequent production cycle, and in the following year (2020), a cross-protection between cGPV and nGPV was found to exist. However, further studies are needed to evaluate the protective efficacy of GPV vaccines against SBDS. Interestingly, the phylogenetic analysis of the vp fragment, which included Hoekstra and Palmivax vaccine strains, was related more to the current nGPV sequences than to the classical GPV strains (including vaccine strains such as Piw-82 and GPV486). The same phylogenetic grouping has been observed in previous studies [1,21]. Cross-protection may exist as nGPV and cGPV are genetically very close, and ducks and geese can be infected with both types of virus.

Although nGPV has a very similar nucleotide sequence to cGPV, in our study the nucleotide identity was 96.42% with reference U25749.1, which is a clearly different virus strain/variant as in experimental settings SBDS can be reproduced in ducks only with nGPV. Geese can be infected with nGPV isolated from Pekin ducks, and the clinical symptoms resemble those of Derzsy’s disease, but with slightly lower mortality rates compared to cGPV infection [22]. Likewise, Pekin ducklings can be artificially infected with cGPV isolated from geese, but without the development of SBDS symptoms, which could only be reproduced with nGPV infection [23]. This indicates that both nGPV and cGPV can infect ducks and geese, but there is a high adaptation to these hosts. With the emergence of the new variant of GPV, and its spreading in the world, it is important to identify the genetic changes that are involved in host adaptation.

The capsid proteins (VPs) are responsible for viral tropism and pathogenicity. Vp1 and Vp3 are surface-exposed proteins that represent the major determinant of viral receptor binding and host specificity.

Analysis of the amino acid sequences of protein-coding regions in the nGPV sequences from China revealed 12 and 8 amino acid changes in the Rep and Vp1 protein, respectively [9,19]. A known function of Rep is to modulate viral replication in host cells, including high-affinity DNA binding to GPV replication origin, helicase activity, and replication of double-stranded linear DNA [24]. In our study, we observed a higher rate of mutations in Rep protein than in Vp1 protein. It can be assumed that mutations in Rep are responsible for higher pathogenicity in Pekin ducks, and the emergence of nGPV in this duck breed, but this hypothesis needs further investigation.

In this study, we also observed that the rate of nGPV and DuCV co-infection was 85.7%. DuCV is frequently isolated from ducks together with other viral or bacterial pathogens [14,25]. In two prevalence studies conducted in the areas affected by SBDS in China, the co-infection rate of nGPV and DuCV was found to be 63.53% and 70% in Pekin ducks [4,26]. Interestingly, in the more recent study, nGPV with distinct mutation pattern in Vp3 protein was isolated from feather sacks of ducks suffering from feather shedding syndrome that is caused by DuCV [26]. Recently, the effect of co-infections was studied in detail in experimental settings [27]. It was observed that after the initial lower viral loads, the loads of both nGPV and DuCV in dually infected birds were higher compared to those infected with only one pathogen.

In this study, we did not perform nGPV virus isolation and experimental infection of the ducks. As the co-infection with DuCV was high, we do not know if nGPV is the only etiological agent of SBDS in this particular case. Nevertheless, the literature data on experimental infections with nGPV and the genetic sequence of the nGPV that we obtained is sufficient to prove the infection in Polish Pekin ducks.

To our knowledge, this is the first report of SBDS caused by nGPV in Pekin ducks in Poland and Europe. It should be emphasized that monitoring and sequencing of waterfowl parvoviruses is important for tracking the viral genetic changes that enable adaptation to new species of waterbirds.

## 5. Conclusions

In summary, we have reported the first record of nGPV cases in Pekin duck flocks in Poland, which was responsible for the SBDS outbreak. Morbidity ranged between 15% and 40%, while the mortality rate was 4–6%. Co-infection with DuCV was detected in six out of eight flocks. The complete coding sequence of nGPV was obtained for four isolates, while for the remaining isolates we obtained a fragment of *vp1* gene. Phylogenetic analysis revealed that the Polish sequences clustered together with nGPV sequences from recent SBDS Chinese outbreaks, with a nucleic acid identity of 98.57–99.28%. It seems that nGPV was present in the past in France causing SBDS in mule ducks, but recently a new disease emerged in Pekin ducks. A higher frequency of amino acid mutations was found in the Rep protein than in Vp1 protein, indicating that the Rep protein might be involved in the pathogenicity of nGPV in Pekin ducks.

## Figures and Tables

**Figure 1 animals-10-02397-f001:**
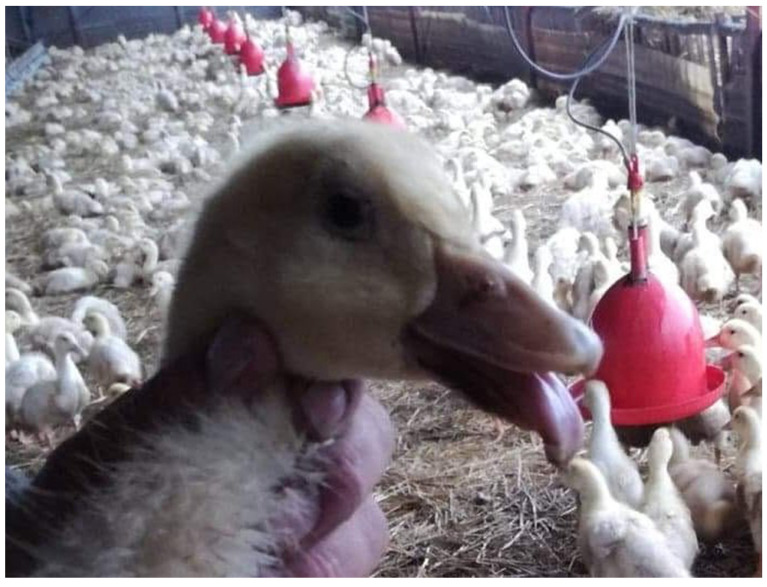
Picture of duck with SBDS signs: shortening of the beak and protruding and dropping tongue.

**Figure 2 animals-10-02397-f002:**
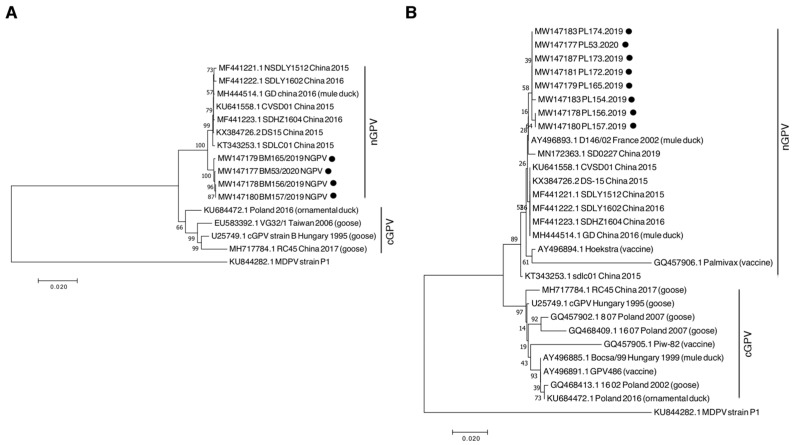
Phylogenetic tree generated based on four complete coding sequences (**A**) and partial *vp1* gene (**B**) with the sequences of other waterfowl parvovirus isolates available in the GenBank database. The Polish sequences obtained from this study are indicated with black dots. The viral sequences isolated from Pekin ducks are shown without brackets. MDPV, Muscovy duck parvovirus; cGPV, classical goose parvovirus; nGPV, novel goose parvovirus. The phylogenetic tree was constructed using the neighbor-joining algorithm with 1000 bootstrap replicates. Bootstrap values are shown on the tree.

**Table 1 animals-10-02397-t001:** List of flocks used in the study with age description, PCR results, and Genbank submission numbers.

No.	Number	Age of Ducks (Weeks)	GPV qPCR/Ct Value	DuCV PCR	Complete Coding Region Genbank Number	P5 Fragment Genbank Number
1	PL174.2019	4	positive/24	negative		MW147183
2	PL172.2019	3	positive/24	positive		MW147181
3	PL173.2019	4.5	positive/29	positive		MW147187
4	PL154.2019	6	positive/32	positive		MW147182
5	PL156.2019	5	positive/26	positive	MW147178	
6	PL157.2019	3	positive/25	positive	MW147180	
7	PL165.2019	4.5	positive/26	positive	MW147179	
8	PL53.2020	4.5	positive/23	n.d.	MW147177	

n.d. = not done.

**Table 2 animals-10-02397-t002:** Amino acid substitutions in Vp1 protein.

Sequence\sAa Position	28	47	89	92	114	116	142	144	149	180	450	498	521	529	558	593	615	660
cGPV (U25749)	Q	K	Q	K	D	Q	D	V	A	A	S	S	Y	L	E	D	G	H
cGPV PL 2016 (KU684472)	H	R	L	K	D	H	E	I	T	V	S	S	H	I	D	E	W	H
nGPV CH 2016 (KX384726)	Q	R	L	K	H	H	E	I	A	V	N	N	H	I	D	D	W	N
nGPV PL 2019 this study	Q	R	L	R	D	H	E	I	A	A	N	N	H	I	D	D/E	W	H

Unique mutations in the Polish nGPV strains are marked in yellow.

**Table 3 animals-10-02397-t003:** Amino acid substitutions in Vp1 Rep protein.

Sequence\Aa Position	22	50	73	131	140	200	211	250	318	350	458	468	497	498	535	549	553	555	564	573	575	576	594	605	609	617
cGPV (U25749)	S	I	Q	K	A	T	A	E	L	A	P	V	P	E	S	P	R	N	T	E	M	E	D	K	N	V
cGPV PL 2016 (KU684472)	P	I	R	K	S	S	V	D	L	A	P	V	P	E	S	R	R	N	S	E	M	E	D	K	T	V
nGPV CH 2016 (KX384726)	P	T	Q	R	S	T	A	D	L	V	P	I	P	R	S	P	K	T	T	Q	M	E	Y	K	D	A
nGPV PL 2019 this study	P	T	Q	R	S	T	A	D	L/R	V	S	I	Q	K	P	P	K	T	T	K	I	D	Y	T	D	A

Unique mutations in the Polish nGPV strains are marked in yellow.

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
