# Peer review of "Short Beak and Dwarfism Syndrome in Ducks in Poland Caused by Novel Goose Parvovirus"

_animals, 2020, doi:10.3390/ani10122397_

Round 1
Reviewer 1 Report
The study reports the sequences of duck parvoviruses found in association with short beak and dwarfism syndrome(SBDS), a disease of farmed ducks. One of the suspected etiological agents of SBDS is novel goose parvovirus (nGPV), a parvovirus closely related to classical goose parvovirus (cGPV). It is thought possible that a circovirus (Duck circovirus - DuCV) is involved as well. The authors provide the first report that nGPV is found in association with SBDS in Europe, supporting their conclusions with phylogenetic analysis. The approach taken is appropriate and the analysis appears to have been performed rigorously. The conclusions seem justified - there is much about SBDS that remains unresolved, but studies like this are absolutely necessary to progress. Minor corrections Lines 14-15 - suggest changing to "Since 2015, outbreaks have occurred in many parts of China, causing economic losses...." Line 78: "the leader in poultry production - "a leader" might be better? Line 255 - "a higher rate of amino acid mutations" - I think the authors probably mean "a higher frequency"Author Response
We thank the reviewer for his or her work. The language correction have been made and marked with green font.
"Minor corrections Lines 14-15 - suggest changing to "Since 2015, outbreaks have occurred in many parts of China, causing economic losses...." Line 78: "the leader in poultry production - "a leader" might be better? Line 255 - "a higher rate of amino acid mutations" - I think the authors probably mean "a higher frequency""
These changes have been put into R1 version of the manuscript.
Reviewer 2 Report
In this MS, the authors firstly reported the nGPV infection being responsible for the SBDS outbreak in Pekin ducks in Poland and Europe. The results provided some valuable information for understanding the molecular epidemiology of nGPV. However, a few modifications are necessary to improve the manuscript.
- Please use three-wire meter in Table 2.
- In line 255-257, the sentence should be “A higher rate of amino acid mutations 255 was found in the Rep protein than in Vp1 protein, indicating that the Rep protein might be involved in the pathogenicity of nGPV in Pekin ducks.
- The pathogenicity of the co-infection of nGPV and DuCV has recently been reported (Yang et al., Poultry Science, 2020, 99(9): 4227-4234.). In the discussion section, the authors should add the reference.
Author Response
We thank the reviewer for his/her work.
- We do not understand what three-wire meter means? Should the amino acid names be written in three letter abbreviation? That would make the table too large. We can change it three letter table if it is what reviewer expect.
- The sentence was changed, changes are indicated in blue in resubmitted manuscript. A frequency world was asked by the other reviewer. Now the sentence is :
A higher frequency of amino acid mutations was found in the Rep protein than in Vp1 protein, indicating that the Rep protein might be involved in the pathogenicity of nGPV in Pekin ducks.
-
The relevant publication has been added and discussed:
In two prevalence studies conducted in the areas affected by SBDS in China, the co-infection rate of nGPV and DuCV was found to be 63.53% and 70% in Pekin ducks [4,26]. Interestingly, in the more recent study nGPV with distinct mutation pattern in Vp3 protein, was isolated from feather sacks of ducks suffering from feather shedding syndrome that is caused by DuCV [26].
Reviewer 3 Report
The paper describes the sequence comparison and alignment of the short beak and dwarfism syndrome (SBDS) virus obtained from diseases in ducks in Poland. The sequence analysis revealed that the virus is distinct from the previously reported isolates from Asia, suggesting that it is a genetic variant of goose parvovirus (GPV). As the first report of a variant GPV from Europe, the study provides interesting insights into the changes in the viral genome probably required for adaptation to the new species of waterfowls in Europe.
While the clinical signs of the disease outbreak and morbidities of 15-40% and mortalities of 4-6% were described, no transmission or experimental infections with the isolates have been carried out in the study, to confirm the etiology, especially as a significant part of the cases were coinfected duck circovirus. It will be important to mention this caveat of the study.
Nevertheless, sequence comparison provides the information on the etiology and the distinct nature of the variant.
Author Response
We thank the reviewer for his or her work.
Reviewer comment :
"While the clinical signs of the disease outbreak and morbidities of 15-40% and mortalities of 4-6% were described, no transmission or experimental infections with the isolates have been carried out in the study, to confirm the etiology, especially as a significant part of the cases were coinfected duck circovirus. It will be important to mention this caveat of the study."
In this study we did not grow the virus on SPF embryonated duck eggs and we did not have performed experimental duck infection to confirm the etiology. We though that we need to publish the results quickly, and also animal experiments policy in Poland and EU limit animal experiments. We think that genetic sequence and the clinical signs are sufficient to prove the infection with this new GPV variant .
We put the sentence to the discussion sequence, marked in red in the manuscript:
"In this study we did not perform nGPV virus isolation and experimental infection of the ducks. As the co-infection with DuCV was high, we do not know if nGPV is the only etiological agent of SBDS in this particular case. Nevertheless, the literature data on experimental infections with nGPV and the genetic sequence of the nGPV that we obtained is sufficient to prove the infection in Polish Pekin ducks."